Long-term monitoring projects of Brazilian marine and coastal ecosystems

Cordeiro Cesar A.M.M. cammcordeiro@pq.uenf.br 1
Aued Anaide W. 2
Barros Francisco 3
Bastos Alex C. 4
Bender Mariana 5
Mendes Thiago C. 6 7
Creed Joel C. 8
Cruz Igor C.S. 9
Dias Murilo S. 10
Fernandes Lohengrin D.A. 11
Coutinho Ricardo 11
Gonçalves José E.A. 11
Floeter Sergio R. 12
Mello-Fonseca Juliana 6
Freire Andrea S. 13
Gherardi Douglas F.M. 14
Gomes Luiz E.O. 15
Lacerda Fabíola 16
Martins Rodrigo L. 17
Longo Guilherme O. 18
Mazzuco Ana Carolina 15
Menezes Rafael 11
Muelbert José H. 19
Paranhos Rodolfo 20
Quimbayo Juan P. 21
Valentin Jean L. 20
Ferreira Carlos E.L. 6
1 PELD Ilhas Oceânicas Brasileiras, Laboratório de Ciências Ambientais, Universidade Estadual do Norte Fluminense , Campos dos Goytacazes , Rio de Janeiro , Brazil
2 PELD Ilhas Oceânicas Brasileiras, Memorial University of Newfoundland , St John’s , Newfoundland , Canada
3 Laboratório de Ecologia Bentônica, IBIO & CIEnAM & INCT IN-TREE, Universidade Federal da Bahia , Salvador , Bahia , Brazil
4 PELD Abrolhos, Departamento de Oceanografia, Universidade Federal do Espírito Santo , Vitória , Espírito Santo , Brazil
5 PELD Ilhas Oceânicas Brasileiras, Marine Macroecology and Conservation Lab, Universidade Federal de Santa Maria , Santa Maria , Rio Grande do Sul , Brazil
6 PELD Ilhas Oceânicas Brasileiras, Laboratório de Ecologia e Conservação de Ambientes Recitais, Universidade Federal Fluminense , Niterói , Rio de Janeiro , Brazil
7 PELD Ilhas Oceânicas Brasileiras, Instituto do Mar, Universidade Federal de São Paulo , Santos , São Paulo , Brazil
8 Departamento de Ecologia, Instituto de Biologia Roberto Alcântara Gomes, Universidade do Estado do Rio de Janeiro , Rio de Janeiro , Brazil
9 Laboratório de Oceanografia Biológica, Departamento de Oceanografia, Instituto de Geociências da Universidade Federal da Bahia , Salvador , Bahia , Brazil
10 PELD Ilhas Oceânicas Brasileiras, Departamento de Ecologia, Universidade de Brasília , Brasília , Distrito Federal , Brazil
11 PELD Ressurgência de Cabo Frio, Instituto de Estudos do Mar Almirante Paulo Moreira (IEAPM) , Arraial do Cabo , Rio de Janeiro , Brazil
12 PELD Ilhas Oceânicas Brasileiras, Marine Macroecology and Biogeography Lab, Universidade Federal de Santa Catarina , Florianópolis , Santa Catarina , Brazil
13 PELD Ilhas Oceânicas Brasileiras, Laboratório de Crustáceos e Plâncton, Universidade Federal de Santa Catarina , Florianópolis , Santa Catarina , Brazil
14 PELD Ilhas Oceânicas Brasileiras, Laboratory of Ocean and Atmosphere Studies (LOA), Earth Observation and Geoinformatics Division, National Institute for Space Research (INPE) , São José dos Campos , São Paulo , Brazil
15 PELD Habitats Costeiros do Espírito Santo, Grupo de Ecologia Bêntica, Departamento de Oceanografia e Ecologia, Universidade Federal do Espírito Santo , Vitória , Espírito Santo , Brazil
16 Conselho Nacional de Desenvolvimento Científico e Tecnológico (CNPq) , Brasília , Distrito Federal , Brazil
17 PELD Restingas e Lagoas Costeiras do norte do Estado do Rio de Janeiro, Instituto de Biodiversidade e Sustentabilidade (NUPEM), Universidade Federal do Rio de Janeiro , Macaé , Rio de Janeiro , Brazil
18 PELD Ilhas Oceânicas Brasileiras, Laboratório de Ecologia Marinha, Departamento de Oceanografia e Limnologia, Universidade Federal do Rio Grande do Norte , Natal , Rio Grande do Norte , Brazil
19 PELD Estuário da Lagoa dos Patos e Costa Marinha Adjacente, Instituto de Oceanografia, Universidade Federal do Rio Grande , Rio Grande , Rio Grande do Sul , Brazil
20 PELD Baía de Guanabara, Instituto de Biologia, Universidade Federal do Rio de Janeiro , Rio de Janeiro , Brazil
21 PELD Ilhas Oceânicas Brasileiras, Centro de Biologia Marinha, Universidade de São Paulo , São Sebastião , São Paulo , Brazil
Meraj Gowhar
Electronic publication date: 2022 Nov 9
Publication date: 2022
Volume: 10
Electronic Location ID: e14313
Received 2022 Jun 20; Accepted 2022 Oct 6
Copyright: ©2022 Cordeiro et al.
Copyright year: 2022
Copyright holder: Cordeiro et al.
License: This is an open access article distributed under the terms of the Creative Commons Attribution License, which permits unrestricted use, distribution, reproduction and adaptation in any medium and for any purpose provided that it is properly attributed. For attribution, the original author(s), title, publication source (PeerJ) and either DOI or URL of the article must be cited.
License URL: https://creativecommons.org/licenses/by/4.0/

Keywords: Ecology, Ocean decade, ILTER, Conservation, Ecosystem management

Funding: “Workshop for Integration of Time Series from Brazilian Marine Ecosystems—Long-Term Ecological Program (PELD)” was supported by the Fundação de Amparo à Pesquisa do Rio de Janeiro E-26/010.000839/2019 Instituto de Estudos do Mar Almirante Paulo Moreira (IEAPM) logistic support Conselho Nacional de Desenvolvimento Científico e Tecnológico (CNPq) Brazilian National Council for Scientific and Technological Development (CNPq; 310517/2019–2) Fundação de Amparo à Pesquisa do Estado do Rio de Janeiro (FAPERJ) E-26/202.310/2019 E-26/202.372/2021 São Paulo Research Foundation FAPESP 2018/21380-0 2021/09279-4 CNPq 304.004/2018-9 APERJ E-26/201.026/2022 This first “Workshop for Integration of Time Series from Brazilian Marine Ecosystems—Long-Term Ecological Program (PELD)” was supported by the Fundação de Amparo à Pesquisa do Rio de Janeiro (FAPERJ, grant E-26/010.000839/2019 CELF-PI), with the Instituto de Estudos do Mar Almirante Paulo Moreira (IEAPM) logistic support. PELD projects listed here were supported by the Conselho Nacional de Desenvolvimento Científico e Tecnológico (CNPq). Guilherme Longo was supported with a research productivity scholarship provided by the Brazilian National Council for Scientific and Technological Development (CNPq; 310517/2019–2). Cesar Cordeiro and Thiago Mendes were supported by the Fundação de Amparo à Pesquisa do Estado do Rio de Janeiro (FAPERJ) with post-doctoral scholarships (grants E-26/202.310/2019 and E-26/202.372/2021). Juan Quimbayo was supported by the São Paulo Research Foundation (FAPESP 2018/21380-0 and 2021/09279-4). Carlos E.L. Ferreira is supported by CNPq (304.004/2018-9) and FAPERJ (E-26/201.026/2022). The funders had no role in study design, data collection and analysis, decision to publish, or preparation of the manuscript.

==============================
Biodiversity assessment is a mandatory task for sustainable and adaptive management for the next decade, and long-term ecological monitoring programs are a cornerstone for understanding changes in ecosystems. The Brazilian Long-Term Ecological Research Program (PELD) is an integrated effort model supported by public funds that finance ecological studies at 34 locations. By interviewing and compiling data from project coordinators, we assessed monitoring efforts, targeting biological groups and scientific production from nine PELD projects encompassing coastal lagoons to mesophotic reefs and oceanic islands. Reef environments and fish groups were the most often studied within the long-term projects. PELD projects covered priority areas for conservation but missed sensitive areas close to large cities, as well as underrepresenting ecosystems on the North and Northeast Brazilian coast. Long-term monitoring projects in marine and coastal environments in Brazil are recent (<5 years), not yet integrated as a network, but scientifically productive with considerable relevance for academic and human resources training. Scientific production increased exponentially with project age, despite interruption and shortage of funding during their history. From our diagnosis, we recommend some actions to fill in observed gaps, such as: enhancing projects’ collaboration and integration; focusing on priority regions for new projects; broadening the scope of monitored variables; and, maintenance of funding for existing projects.

Introduction

Ecological monitoring can provide valuable information for the management and sustainability of ecosystems, including our survival in the face of increasing changes in ecosystem functioning and degradation of ecological services (Lindenmayer & Likens, 2009). Such biodiversity monitoring is paramount in order to act before and while changes are still manageable, by implementing thresholds and warning systems, guiding restoration, and building efficient natural observatories (Canonico et al., 2019). Thus, reliable data from monitoring efforts depends on optimal planning and data management to maximize its usage and ensure its longevity (Magnusson et al., 2013; Canonico et al., 2019).

Long-term ecological monitoring is an essential tool to raise red flags about declining populations, changes in species distribution, structure and stability of food webs, increase in functional vulnerability and risk of species extinction (Gaiser et al., 2020; Lindenmayer & Likens, 2009), helping to track ecosystem shifts. Long-term monitoring is also essential to understand ecosystem dynamics and ecological processes at different levels of organization and across scales (Norström et al., 2009; Cruz et al., 2015), as well as a proving ground to evaluate marine protected areas (MPAs) effectiveness (Roos et al., 2020) and forecast changes in ecosystems (Capitani et al., 2021). Globally, monitoring efforts are mostly associated with Northern Hemisphere biomes following the scientific historical background of economically developed countries (Mirtl et al., 2018; Muelbert et al., 2019). Therefore, the Global South should be considered a priority region for establishing monitoring projects due to its high biodiversity, ongoing degradation and sparse long-term initiatives.

The Brazilian Long-Term Ecological Research Program (Programa de Pesquisas Ecológicas de Longa Duração—PELD, in Portuguese) was conceived in 1997 to support long-term research and monitoring in Brazil (Barbosa, 2013) and is a national effort funded by the Brazilian government and coordinated by the CNPq (Conselho Nacional de Desenvolvimento Científico e Tecnológico, in Portuguese). PELD is a member of the ILTER (International Long-Term Ecological Research) network encompassing 44 countries and 700 research sites in a variety of ecosystems across the planet (Mirtl et al., 2018), of which 63 are coastal and 52 are marine sites (Muelbert et al., 2019; Muelbert et al., 2020). Currently, there are 34 active PELD sites in Brazilian territory, nine of which are in marine or coastal habitats (Brito, De Oliveira & Mamede, 2020).

As we enter the United Nations Decade of Ocean Science for Sustainable Development (2021–2030 Ocean Decade), clean, accessible, and resilient oceans are key to achieving Sustainable Development Goals (Heymans et al., 2020; Visbeck, 2018). Brazil’s role in monitoring marine and coastal ecosystems is crucial since its jurisdictional waters represent a large part of the South Atlantic Ocean, and harbor unique ecosystems such as the Abrolhos Bank reef complex (Leão & Kikuchi, 2001), the Rocas Atoll (Longo et al., 2015), and the Great Amazon Reef (Moura et al., 2016). In early 2020, experts involved in monitoring marine and coastal ecosystems from PELD projects took part in a workshop to compile the information and scientific production generated by those monitoring nuclei. The meeting aimed to create a “state of the art” of those monitoring sites and build a framework for future combined efforts and collaboration. This manuscript is one of the products of the first “Workshop for Integration of Time Series from Brazilian Marine Ecosystems—Long-Term Ecological Program (PELD)”. Using an expert group workflow, we investigated the main characteristics of long-term monitoring projects established in Brazilian coastal and marine environments to highlight their main scientific outputs, funding issues and human resources, as well as the spatial and temporal scales covered by these initiatives. From that picture, we point out spatial gaps and collaboration strengths to help frame future directions for the long-term monitoring of marine and coastal ecosystems in Brazil. Furthermore, we show a picture of the largest country in the Global South, representing a considerable part of the southern Atlantic shores within a single national jurisdiction, which may benefit other countries with similar challenges in monitoring marine and coastal areas.

Material and Methods

Coordinators of PELD projects from coastal and marine areas (until 2019) and experts in marine biological monitoring from Brazilian universities were invited to participate in the workshop. We did not include individual monitoring efforts such as programs supported by oil companies and other mixed-funds sources due to their major focus on umbrella species. We restricted our data to research groups linked to PELD projects to evaluate the functioning of a national-level funding program with expansion potential. In total, twenty-eight researchers from 16 universities and research institutes from eight states and the Federal District attended the workshop. These researchers contributed with data from nine projects of PELD program (Table S1) which had completed more than 5 years of uninterrupted sampling, irrespective of frequency (annual, monthly, weekly etc.). The workshop was divided into three phases:

1. Identification of the temporal and spatial scales of the monitoring programs, and their monitored variables. This part aimed at making a diagnosis of the overlapping targets of monitoring projects and their history;

2. Evaluation of possible approaches to the analysis of time series. Here, we aimed at investigating which broad ecological questions and challenges associated with climatic changes can be tackled in this conjunct effort;

3. Implementation of strategies for the execution of scientific deliverables from the existing data. Lastly, we aimed at proposing coordinated actions for sustainable management of marine areas in Brazil based on the previous outputs, strengths, and gaps of current monitoring knowledge in an integrated effort with environmental agencies and stakeholders from monitoring locations.

The following indicates the results of the workshop’s first phase and they were based on information collected from coordinators and collaborators of PELD projects regarding temporal and spatial scales over which they operate. Phases 2 and 3 are still in development and will not be treated here. Data from each project was obtained by direct structured interviews with coordinators and complemented by answering digital forms (Google Forms®) sent after the meeting (Table S2). The questions addressed involved (1) details of each PELD project, including the number of researchers, students and partner institutions involved, (2) the biological and physical components monitored by each project, (3) funding information, and (4) scientific contribution and human resources formation. The complete questionnaire with all evaluated variables can be found in the Supplementary Material (Table S2). We did not evaluate differences in methods applied for monitoring the variables because that varied considerably among projects and, sometimes, methods changed within the same project throughout its history.

We applied generalized linear models (GLMs) to assess the relationship between scientific production and project monitoring time using log as a link function to adjust to the Poisson distribution of scientific production. GLMs were run using the glm function from the stats package in R software (R Core Team, 2020). We built a network of projects partners based on direct relationships between institutions associated with projects as indicated by coordinators and collaborators in the digital forms. From the network, we calculated four global metrics based on an undirected network: transitivity, modularity, connectance, and centralization. Transitivity measures the probability that the adjacent vertices of a node are connected (Wasserman & Faust, 1994). Modularity describes a compartmentalized distribution of interactions among projects (e.g., Olesen et al., 2007). Connectance is the proportion of realized links relative to all the possible links in the network (Boccaletti et al., 2006), and centralization represents the heterogeneity in the distribution of institutions and projects. Associations were evaluated using the igraph package (Csardi & Nepusz, 2006), and graphs were plotted using ggplot2 package (Wickham, 2016).

Results

Twelve monitoring locations from nine projects (some PELD projects monitor more than one site, such as PELD ILOC which monitors four sites) within the PELD program were identified along ∼3,200 km of the Brazilian coast, from Pernambuco to Rio Grande do Sul (Fig. 1). A 2,000 km gap was found in the north and northeast coastal regions (from Amapá to Rio Grande do Norte), and another 1,000 km gap was observed in the southeast region (from São Paulo to Rio Grande do Sul; Fig. 1). Habitats monitored by the research groups mainly included subtidal ecosystems, while terrestrial elements (coastal vegetation and fauna) were studied only in two programs (Table S1). Shallow areas (<50 m deep) with reef environments (biogenic and non-biogenic) were the most frequently and widespread surveyed habitats (six of nine projects, Table S1). The distribution of reef environments included both tropical and subtropical domains and all Brazilian oceanic islands. Unconsolidated substrate habitats were monitored in four out of nine projects (Table S1) in shallow coastal habitats. All evaluated PELD projects lie within or in the vicinity of MPAs (Table S1).

Figure 1 (A) Map indicating the monitored sites, (B) length of time series (continuous lines before bars indicate monitoring period before PELD support), and (C) general indicators monitored at each long-term monitoring site in the Brazilian coast.

The list of monitored variables can be found in Table S5 in the Supplementary Material. Polygons in dark blue (A) indicate priority areas for conservation according to Magris et al. (2021). ‘others’ may include isotopes, carbon stock, reef accretion and in situ primary production. ILOC, Monitoramento de Longa Duração das Comunidades Recifais das Ilhas Oceânicas Brasileiras; TAMS, Tamandaré Sustentável; CCAL, Costa dos Corais Alagoas; HCES, Habitats Costeiros do Espírito Santo; RLaC, Restingas e Lagoas Costeiras do norte do Estado do Rio de Janeiro; RECA, Ressurgência de Cabo Frio; PEBG, Estrutura e Funções do ecossistema da Baía de Guanabara; ELPA, Estuário da Lagoa do Patos e Costa Marinha Adjacente.

Monitoring targets varied among projects and represented different levels of organization (Table S5) such as communities (e.g., benthic community, zooplankton), assemblages (e.g., ichthyofauna), and key taxa locally monitored by some PELD projects (e.g., the structuring coastal vegetation Clusia sp., and the scleractinian coral Montastraea cavernosa). Reef fish were the biological indicator with the greatest spatial extent in monitoring, covering tropical and subtropical domains (Fig. 1, Table S3), and monitored in 89% of PELD locations. The aquatic flora (macroalgae and/or seagrass) and sessile invertebrates were frequent groups in monitoring initiatives (67%). The coastal vegetation (mangrove forests and/or restinga sand vegetation) and the mobile invertebrate fauna were monitored sparsely along the coast. Neritic components (e.g., phytoplankton, zooplankton, marine mammals) were mostly monitored at southern locations, while socio-economic indicators were monitored at northeastern sites (i.e., CCAL, Abrolhos and TAMS; Fig. 1, Table S5).

Most of the PELD monitoring projects had short temporal data sets (median = 12 years, Q1 = 6, Q3 = 22), with the longest time associated with the Lagoa dos Patos PELD (since 1993, 29 years). The monitoring carried out by the Brazilian Navy’s Sea Studies Institute, the Instituto de Estudos do Mar Almirante Paulo Moreira (IEAPM) in Arraial do Cabo, which started in 1974, was funded as a PELD site in 2016. All sites had, at least, an annual monitoring frequency since the start of projects (Fig. 1, Table S1), although some had a greater frequency during part of the monitoring period (weekly = 11.1%, monthly = 44.4% or quarterly = 66.6%, during a couple of years). The plankton-associated groups had the longest monitoring time, especially associated with the oldest locations (ELPA = 29 years, and PEBG = 25 years). ‘Others’ consisted of temporal monitoring describing geochemical processes, multi taxa approaches, stable isotopes, water temperature, reef accretion, carbon stocks and geological aspects which are particular to some PELD locations (Fig. 1).

The federal agency Conselho Nacional de Desenvolvimento Científico e Tecnológico (CNPq) was the main funding source of PELD projects, followed by state level public funding agencies and the National Education Agency—CAPES (Coordenação de Aperfeiçoamento de Pessoal de Nível Superior in Portuguese). The latter, usually supported programs by supplying Ph.D. and M.Sc. scholarships. The capacity building and training of scientists was a valuable outcome of the programs, accounting for at least 291 PhD and Master’s thesis. The results and actions of marine and coastal PELD projects were communicated as scientific peer-reviewed publications (peer-reviewed articles, book and book chapters; n = 530 until January 2020) which constitute a large part of the results of those monitoring efforts (Table S4) and have increased exponentially since monitoring was set up at each site (Fig. 2). Scientific events, websites and social media were the most used ways to communicate and disseminate the results from PELD projects, whereas large-scale communication outlets such as broadcast television, newspaper and radio were seldom used (Fig. 3).

Figure 2 Relationship between scientific production and monitoring time of long-term monitoring sites (PELD) in Brazilian coastal ecosystems.

Generalized linear model (GLM) adjusted to values by using Poisson distribution and log as link function (p < 0.001, AIC = 386.7). Shaded area are 95% confidence intervals predicted by GLM. ILOC, Monitoramento de Longa Duração das Comunidades Recifais das Ilhas Oceânicas Brasileiras; TAMS, Tamandaré Sustentável; CCAL, Costa dos Corais Alagoas; HCES, Habitats Costeiros do Espírito Santo; RLaC, Restingas e Lagoas Costeiras do norte do Estado do Rio de Janeiro; RECA, Ressurgência de Cabo Frio; PEBG, Estrutura e Funções do ecossistema da Baía de Guanabara; ELPA, Estuário da Lagoa do Patos e Costa Marinha Adjacente.

Figure 3 Accumulated funding received (A) and frequency of communication outlets used to inform results (B) of long-term monitoring projects (PELD) in Brazil.

National level funding agencies: CAPES, Coordenação de Aperfeiçoamento de Pessoal de Nível Superior; CNPq, Conselho Nacional de Desenvolvimento Cientíûco e Tecnolôgico; FAPs 3 state level public funding agencies; always = 100% of time, never = 0 % of time, sometimes = 50% of the time. ILOC, Monitoramento de Longa Duração das Comunidades Recifais das Ilhas Oceânicas Brasileiras, TAMS 3 Tamandaré Sustentável; CCAL, Costa dos Corais Alagoas; HCES, Habitats Costeiros do Espírito Santo; RLaC, Restingas e Lagoas Costeiras do norte do Estado do Rio de Janeiro, RECA 3 Ressurgência de Cabo Frio; PEBG, Estrutura e Funções do ecossistema da Baía de Guanabara, ELPA 3 Estuário da Lagoa do Patos e Costa Marinha Adjacente.

We observed a modular PELD projects network (modularity = 0.18), which was formed by few and poorly connected groups (mean distance between nodes = 3.2) with low centralization (0.31) and low transitivity (0.01). Furthermore, we observed few connections between groups of nodes (triads), i.e., institutions. The PELDs ELPA (n = 20), RLaC (n = 13) and PEBG (N = 11) had the highest number of direct connections (i.e., degree), but 61.4% were with single nodes. The largest subgroup detected was formed by Universidade Federal do Ceará (UFC), Instituto Chico Mendes de Conservação da Biodiversidade (ICMBio), PELD-ELPA and PELD-TAMS, indicating the low number of shared vertices among network components. A maximum of four nodes shared between PELD nuclei was observed, despite being located along the coast and some sharing the same target habitats (Fig. 4). The Universidade Federal Fluminense (UFF), ICMBio and Universidade de São Paulo (USP) were all connected to five PELD sites each. Governmental institutions accounted for only 12.5% of nodes and were more associated with northeastern PELD sites. Older projects (>20 years) had more international collaborators (75% of all) but shared few connections with other PELD sites (Fig. 4). NGOs were connected to only two nodes (4%) of all connections.

Figure 4 Network of institutions involved with marine and coastal long-term monitoring projects (PELD) in Brazil.

ILOC, Monitoramento de Longa Duração das Comunidades Recifais das Ilhas Oceânicas Brasileiras; TAMS, Tamandaré Sustentável; CCAL, Costa dos Corais Alagoas; HCES, Habitats Costeiros do Espírito Santo; RLaC, Restingas e Lagoas Costeiras do norte do Estado do Rio de Janeiro; RECA, Ressurgência de Cabo Frio; PEBG, Estrutura e Funções do ecossistema da Baía de Guanabara; ELPA, Estuário da Lagoa do Patos e Costa Marinha Adjacente. Numbers represent different partner institutions, see Table S3 for details.

Discussion

Long-term monitoring projects in the Brazilian marine environment are recent, poorly connected, but scientifically productive with an important contribution to academic and human resources training. Long-term ecological monitoring programs seek to give the planned return after several years of existence (Caughlan & Oakley, 2001) because the questions raised by this model of ecological research require long time series to allow temporal patterns to emerge (Giron-Nava et al., 2017). The exponential growth in the scientific production of PELD projects was not surprising as it has also been observed for terrestrial LTER (Randig, 2019). After the settling phase, established experience, network and infrastructure give support for knowledge building and the formation of qualified human resources. Thus, efforts and investment in such programs have a key role and contribution to scientific advancement, with a legacy of human resources and networking, which will boost policymaking outcomes (Hughes et al., 2017).

Changes in global environmental stability and their effects on biodiversity and ecosystem services are the main concerns of environmental policy (Haase et al., 2018) and monitoring is a key objective. Long-term monitoring programs are essential to detect changes associated with anthropogenic stressors and disentangle the real effects from ‘background noise’ enabling predictive approaches to ecosystem change (Magurran et al., 2010). Monitoring of Essential Biological (EBVs) and Oceanographic Variables (EOVs) has been one of the aims of global strategies focused on the United Nations Sustainable Development Goals (Kissling et al., 2018). Variables monitored by the PELD projects, such as water temperature, fish and plankton abundance, and macroalgae coverage are within those EOVs and EBVs (Miloslavich et al., 2018). Despite the short time-series of most PELD projects, the monitoring of these key variables at a large spatial scale is advantageous for complying with emerging international efforts on biodiversity monitoring and best practices in globally integrated observing systems.

PELD projects included most coastal environments along the Brazilian coast, with monitoring sites targeting different habitats and taxa. Asymmetries in the number of monitored variables and habitats indicated older and more diverse initiatives concentrated on the southern Brazilian coast, where more traditional universities and institutes are established. There are gaps related to estuaries and intertidal areas’ representativeness, especially in the North and Northeast regions, leaving aside one of the most extensive mangrove forests in the world (Diniz et al., 2019). Additionally, the absence of offshore and underrepresentation of intertidal and unconsolidated substrate habitats could also stimulate joint efforts to build the ground for such monitoring. On the other hand, reef habitats were better covered by monitoring initiatives, which resulted in an opportunity to increase the interaction among marine and coastal PELD projects; for instance, by establishing standardized methods and efforts for sampling, and developing integrated large-scale experiments. Also, considering the seascape heterogeneity, conjunct and individual monitoring efforts must seek to design protocols to account for such habitat representativeness to allow observing changes at the ecosystem level and facilitating the integration of large-scale analysis. On that matter, benthic cover data and depth are both crucial to identifying seascape features and are already contemplated by projects monitoring reef habitats.

Most of the PELD sites were associated with MPAs, many of which are conservation priority areas (Magris et al., 2021). The bias of monitoring locations within protected areas is associated with historical relationships with the MPA’s creation based on management demands and previous scientific knowledge of local ecosystem importance. Thus, knowledge about important and threatened ecosystems outside MPAs, for example, located close to urban areas is missing. Only the PEBG project is located in an important mosaic of ecosystems surrounding a densely populated area, the Rio de Janeiro metropolitan region (IBGE, 2018). This indicates poor representation and knowledge gaps in priority areas, raising a red flag for potential targets on the establishment of new monitoring locations and their connectivity. New PELD projects established in 2020 (after the workshop) are now monitoring more important areas in the north (PELD Sistema de Recifes Mesofóticos da Foz do Rio Amazonas—GARS), northeast (PELD Costa Semi-Árida do Brasil, Ceará) and south coasts (PELD Sistema Estuarino de Laguna e Adjacências)). However, the current scenario indicates the need for expanding monitoring sites towards sensitive and underrepresented ecosystems (e.g., mangrove areas in the North region) in a coordinated fashion. Such efforts will also foster the decentralization of monitoring efforts, human resources formation, and financing of related institutions.

Continuous financial support is crucial for long-term ecological monitoring programs (Caughlan & Oakley, 2001) and, indeed, it was the main challenge faced by all initiatives present at the workshop. Project coordinators reported unstable financial support, during which only core activities were preserved. The expansion in groups and variables sampled was possible only during less troubled periods—a frequent problem in long-term monitoring (Caughlan & Oakley, 2001). PELD initiatives are at risk because of the continuous depletion of funds for scientific institutions, and the future scenario is bleak (Hallal, 2021). The integration of efforts with other financing sources such as mixed-sourced grants (e.g., Instituto Serrapilheira, Fundação ‘O Boticário’, FUNBIO) and state-level public funding agencies (i.e., FAPs) may help increase the lifespan of monitoring programs. Also, strengthening and maintaining LTER initiatives may contribute to increasing the odds of consortium-type applications for international grants allowing also widening the scale of questions answered by those programs (Haase et al., 2018). The Ocean Decade and the association of global monitoring programs, such as ILTER and GEO BON (Muelbert et al., 2019), may facilitate the integration of other monitoring efforts and increase the access to new funding. The PELD program is already part of the ILTER scheme and, considering the geographical dimension of Brazil and its ecosystem representativeness, integrated efforts in monitoring marine and coastal ecosystems would help fill a large gap in data in the South Atlantic. Integration efforts should aim at standardizing monitoring protocols and leverage the formation of human resources by providing access to training on best practices of monitoring methodologies, and curating and analyzing data. However, the network analysis indicated that PELD projects have negligible direct interaction mediated by a few institutions, which are not shared by all projects and would not act as a bridge for such integration. This scenario indicates that integration at the national level is the first challenge to be overcome, and the workshop had the role of starting this dialogue since there is no central institution with this vocation. Only PELDs ELPA (Lemos et al., 2022), RLaC (Cardoso et al., 2021) and Abrolhos (Moura, Favero & Teixeira, 2022) have their datasets available in open repositories (GBIF) which makes integration difficult, besides methodological differences. Other projects are still in an embargo period but should allocate their data in open repositories after this period, around 2023, creating a fertile ground for collaborations.

Best practices on future efforts should include several actions indicated throughout the text and, in general, should include: (1) the creation or adoption of standard and detailed protocols for measuring variables (Beard, Scott & Adamson, 1999), preferably following international agreements; (2) detailed recording of methodology and any changes or adaptations made over time; (3) quality assessment and control protocols, including indications of how to deal with issues (e.g., missing or outlier data, taxonomic resolution, calibration procedures etc.—Ellingsen et al., 2017); (4) synchronicity and compatible spatial replicability of measurements (Beard, Scott & Adamson, 1999); (5) continuous refreshment on training and integration of personnel from participant groups for leverage; follow open science protocols to guarantee transparency, credibility, replicability and reuse of data (UNESCO, 2021); and (6) periodic planning reassessment and reevaluation.

Increasing the communication among scientists and monitoring programs is important and to sustain and expand monitoring initiatives. However, the delivery of results and the appropriation of the generated knowledge by society and decision-makers may be the most important aim of all. “The science we need for the ocean we want” is the Oceans Decade motto, and it is a general feeling that it must be taken seriously by those hoping for a sustainable ocean which scientists will be able to monitor for centuries to come. To achieve these goals, our diagnosis supports a few recommendations: (1) enhance integration and collaboration among monitoring projects within the PELD program and other initiatives; (2) support existing sites and creation of new ones in vulnerable and underrepresented regions and ecosystems; (3) enhance the collaboration among PELD projects to exchange best practices, assure coverage of EOVs and EBVs and broaden the scope of indicators monitored; (4) guarantee the maintenance of long-term funding for monitoring Brazilian marine ecosystems.

Supplemental Information

Supplemental Information 1 Supplementary tables

Variables monitored in each site from this study and other information.

Click here for additional data file.

The authors are grateful for the contribution with data and/or insights of Esteves FA, Zalmon IR, Lotufo TMC, Suhett AC, Bernardino AF, Villaça R, Malhado AC, Pires D, Contreras F, Calado L, Calderon EM, Kassuga A, Secchi E, Moura RL, Ferreira BP and Francini-Filho RB.

Additional Information and Declarations

Competing Interests

Author Contributions

Data Availability

Juan P. Quimbayo is a PeerJ Academic Editor. The authors declare there are no competing interests.

Cesar A.M.M. Cordeiro conceived and designed the experiments, performed the experiments, analyzed the data, prepared figures and/or tables, authored or reviewed drafts of the article, and approved the final draft.

Anaide W. Aued performed the experiments, authored or reviewed drafts of the article, and approved the final draft.

Francisco Barros performed the experiments, authored or reviewed drafts of the article, and approved the final draft.

Alex C. Bastos performed the experiments, authored or reviewed drafts of the article, and approved the final draft.

Mariana Bender performed the experiments, authored or reviewed drafts of the article, and approved the final draft.

Thiago C. Mendes performed the experiments, authored or reviewed drafts of the article, and approved the final draft.

Joel C. Creed performed the experiments, authored or reviewed drafts of the article, and approved the final draft.

Igor C.S. Cruz performed the experiments, authored or reviewed drafts of the article, and approved the final draft.

Murilo S. Dias performed the experiments, authored or reviewed drafts of the article, and approved the final draft.

Lohengrin D.A. Fernandes performed the experiments, authored or reviewed drafts of the article, and approved the final draft.

Ricardo Coutinho performed the experiments, authored or reviewed drafts of the article, and approved the final draft.

José E.A. Gonçalves performed the experiments, authored or reviewed drafts of the article, and approved the final draft.

Sergio R. Floeter performed the experiments, authored or reviewed drafts of the article, and approved the final draft.

Juliana Mello-Fonseca performed the experiments, authored or reviewed drafts of the article, and approved the final draft.

Andrea S. Freire performed the experiments, authored or reviewed drafts of the article, and approved the final draft.

Douglas F.M. Gherardi performed the experiments, authored or reviewed drafts of the article, and approved the final draft.

Luiz E.O. Gomes performed the experiments, authored or reviewed drafts of the article, and approved the final draft.

Fabíola Lacerda performed the experiments, authored or reviewed drafts of the article, and approved the final draft.

Rodrigo L. Martins performed the experiments, authored or reviewed drafts of the article, and approved the final draft.

Guilherme O. Longo performed the experiments, authored or reviewed drafts of the article, and approved the final draft.

Ana Carolina Mazzuco performed the experiments, authored or reviewed drafts of the article, and approved the final draft.

Rafael Menezes performed the experiments, authored or reviewed drafts of the article, and approved the final draft.

José H. Muelbert performed the experiments, authored or reviewed drafts of the article, and approved the final draft.

Rodolfo Paranhos performed the experiments, authored or reviewed drafts of the article, and approved the final draft.

Juan P. Quimbayo performed the experiments, authored or reviewed drafts of the article, and approved the final draft.

Jean L. Valentin performed the experiments, authored or reviewed drafts of the article, and approved the final draft.

Carlos E.L. Ferreira conceived and designed the experiments, performed the experiments, authored or reviewed drafts of the article, and approved the final draft.

The following information was supplied regarding data availability:

The data and code are available at Github: https://github.com/cammcordeiro/workshop_temporal.

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
