# Peer review of "Long-term monitoring projects of Brazilian marine and coastal ecosystems"

_PeerJ, doi:10.7717/peerj.14313_

## Round 0.1 · original submission · Major Revisions

Dear authors, as you shall see, the reviewers have commented on your manuscript and are suggesting considerable changes/ improvements in your manuscript. Kindly consider these suggestions and submit your revised manuscript including a rebuttal to their queries and suggestions.

·

Basic reporting

The English is generally of high quality. The few small errors I detected are listed below.
Line 142: “it” should be “they”
Line 153: “was” should be “were”
Line 154: “build” should be “built”
Line 172: “habitat” should be “habitats”
Line 226: “are sought” should be “seek”
Figure 1: MPAs not in legend.
Figure 3B: The always, sometimes, never gradation should be ordinal (never, sometimes, always)
Figure 4: “foreing” should be foreign

The raw data all appears to be present, but Table 4S should be self contained and have the age of the projects.

Experimental design

The aims of the article I believe are within the scope of the journal and will provide important information for planning LTER sites in Brazil.

Validity of the findings

The findings appear to be clear, strong and well analyzed.

Additional comments

Although perhaps not within the scope of the article, I believe it important to consider the scale of sampling to determine whether results will be scalable and not just whether the variables have the same names (the major problem with EBVs). One of the problems with most LTER research is that it does not consider the landscape (seascape) and how it will change in the future. This is especially important in the marine environment where just about everything depends on depth. When landscapes are not sampled in proportion to availability, the loss of a particular feature being studied (e.g. reefs, inter-tidal areas) becomes hard to evaluate in the case of general changes, such as sea level. The question is whether the feature has been lost or just migrated across the seascape. Standardized seascape sampling at least at some scales would help unite the sites and force them to sample where there is presently nothing of particular interest to that team because of present water levels.

Reviewer 2 ·

Basic reporting

Long-term monitoring projects of Brazilian marine and coastal ecosystems (#74675)

It is clear that long-term monitoring is essential if we are to quantify and ameliorate the biodiversity crisis. As the authors of this paper make clear the Global South is woefully underrepresented in these schemes. This contribution thus makes an important step towards filling this gap.

The paper itself reads as descriptive overview of the PELD program in Brazil – and as such is a useful contribution. However, there is relatively little consideration of other initiatives that assemble biodiversity databases – e.g. the LTER scheme, GBiF, BioTIME, RivFISH. Recognising how the PELD initiative slots into the global challenge will increase the relevance of the paper to a wider audience.

The focus of the paper is on the ‘what’, ‘where’ and ‘when’ for the schemes within its terms of reference. But it had little to say about the data these schemes produce. Where are these data held? How are they checked? Will they be preserved for future generations of scientists? And, are they open access? These aspects need more discussion.

More consideration of the extent and challenges of monitoring schemes in other ecosystems and biomes in Brazil would also greatly enhance the general significance of the paper.

It is unclear from the text if these monitoring programmes are for populations or for assemblages or both. Consistency of sampling methods, and ideally effort, is important. Was this evaluated?

I am not convinced that the network diagram – Figure 4 – adds much to the paper. I found it difficult to relate the discussion about Figure 4 to the plot itself. (Note also misspelling of foreign in the figure legend).

The paper needs a table listing the schemes, taxon of interest, duration, brief description of methods, and reference and/or url.

The authors note as a goal the pursuit of best practice – a useful contribution of this paper would be to include a section on what that best practice consists of.

A couple of small typos

83 track ecosystem shifts. They are also essential to – I think the ‘They’ refers to monitoring, so should be ‘It’
154 Team 2020). We build – should be ‘built’

Experimental design

N/A

Validity of the findings

N/A

Additional comments

As I noted this is a very descriptive paper - but it tackles an important theme in ecology (namely the importance of data from under-represented parts of the world). I think they should provide a list of the schemes, and mention whether the data are open access

---

## Round 0.2 · accepted · Accept

Thank you very much for improving upon your manuscript. Best of luck and congratulations again

·

Basic reporting

I have only a few tiny, nonessential suggestions:

Line 158: Delete "It is worth noting"
Line 282: Delete "On that matter"
Line 325: better as "which makes integration difficult"

Experimental design

I have only a few tiny, nonessential suggestions:

Line 158: Delete "It is worth noting"
Line 282: Delete "On that matter"
Line 325: better as "which makes integration difficult"

Validity of the findings

Nothing to add.